# Principal Components Analysis based frameworks for efficient missing data imputation algorithms

## Abstract

Missing data is a commonly occurring problem in practice. Many imputation methods have been developed to fill in the missing entries. However, not all of them can scale to high-dimensional data, especially the multiple imputation techniques. Meanwhile, the data nowadays tending toward high-dimensional. Therefore, in this work, we propose *Principal Component Analysis Imputation* (PCAI), a simple but versatile framework based on Principal Component Analysis (PCA) to speed up the imputation process and alleviate memory issue of many available imputation techniques, without sacrificing the imputation quality in term of MSE. In addition, the frameworks can be used even when some or all of the missing features are categorical, or when the number of missing features is large. Next, we introduce *PCA Imputation - Classification* (PIC), an application of PCAI for classification problem with some adjustment. We validate our approach by experiments on various scenarios, which shows that PCAI and PIC can work with various imputation algorithms, including the state-of-the-art ones and improve the imputation speed significantly, while achieving competitive mean square error/classification accuracy compared to direct imputation (i.e., impute directly on the missing data).

## 1 Introduction

Despite recent efforts in directly handling missing data [1, 2, 3, 4], missing data imputation approaches [5, 6, 7] remain commonly used. This is because directly handling missing data can be complicated and usually are developed for specific target problems or models, while imputation can be more versatile. Specifically, an important advantage of imputation is that the imputed data becomes *complete*, i.e., no longer have any missing values. Therefore, it is easier to continue with other preprocessing steps, analysis, and data visualizations. Furthermore, one can deploy many models and choose the best one by using the available packages or software tools for non-missing data. Meanwhile, directly handling missing data strategies do not have these advantages. They are more complicated and not that readily available.

Many techniques have been developed for missing data imputation, ranging from traditional techniques such as MICE [5], K-Nearest Neighbors to recent machine learning/deep learning techniques such as GAIN [6], DL-GSA [7]. However, most of them are computationally expensive for big datasets. For example, experiments in [8] show that under their experiment settings, for Fashion MNIST [9], a dataset of 70,000 samples and 784 features, the MICE [5] and missForest [10] techniques are unable to finish the imputation process within three hours for a missing rate (the ratio between the number of missing entries versus the total number entries in the dataset) of 20%. Since datasets nowadays are trending towards larger sizes [11], with hundreds of thousands of features [12], it is crucial to speed up the available imputation techniques. Taking into account resource

consumption and availability such speed up cannot be achieved by only providing more and better hardware but by the development of new methods.

To achieve this goal, this work introduces two novel frameworks based on Principal Component Analysis (PCA) to speed up the imputation process of many available techniques or the imputation-classification process for missing data classification problems. The first framework, **PCA Imputation (PCAI)** is proposed to speed up the imputation speed by partitioning the data into the fully observed features partition and the partition of features with missing data. After that, the imputation of the missing part is performed based on the union of the PCA - reduced version of the fully observed part and the missing part. Interestingly, it turns out that the method has a great potential to aid the performance of methods that rely on many parameters, such as Deep Learning imputation techniques. Meanwhile, the second one, **PCA Imputation - Classification (PIC)** is proposed to deal with the missing data classification problems where dimension reduction is desirable in advance of the model training step. PIC is based on PCAI with some modifications. Note that these frameworks are different from the methods developed for principal component analysis under missing data presented in [13, 14], which are about how to conduct PCA when the data contains missing values.

In summary, the contributions of this article are: (i) we introduce **PCAI** to improve the imputation speed of many available imputation techniques; (ii) we introduce **PIC** to deal with missing data classification problems where dimension reduction is desirable; (iii) we analyze the potential strength and drawbacks of these approaches; and (iv) we illustrate via experiments that our frameworks can work with various imputation strategies while achieve comparable or even lower mean square error/higher classification accuracies compared to the corresponding original approaches, and alleviate the memory issue in some approaches.

The rest of the paper is organized as follows. In Section 2 and Section 3, we review some related work in the field of missing data, and review two popular formulations of PCA. Next, in Section 4, Section 5, and Section 6, we introduce our novel PCAI and PIC frameworks, and study their relation to previous works, respectively. After that, in Section 7, we demonstrate their capabilities via experiments on various datasets. The paper ends with conclusions, remarks, and future works in Section 8.

## 2    Related Works

Various works have been published on missing data imputation to deal with different data analysis situations. As an example, if one is interested in modeling the uncertainty associated with the imputation, suitable approaches can be multiple or Bayesian imputation techniques such as multiple imputations using Deep Denoising Autoencoders [15], Bayesian Principal Component Analysis-based imputation [16], and extreme learning machine multiple imputation [17]. In addition, graphical models can be prominent candidates when transparency, estimability, and testability are desirable, and these approaches can provide meaningful performance guarantees even if the missing values are not at random [18]. Next, for continuous data, matrix completion techniques such as Fast Alternating Least Squares [19], softImpute [20] can quickly give good results. In biology, the missing values are often categorical, and the imputed values need to be interpretable. In such cases, classification techniques or tree-based methods such as decision trees and fuzzy clustering with iterative learning (DIFC) [21], missForest [10], the DMI algorithm [22], and sequential regression trees [23] are well-suited. In addition, some recently developed methods that can handle mixed missing data are SICE [24], FEMI [25], and HCMM-LD [26]. When the sample sizes are large enough compared to the number of features, deep learning techniques such as Multiple Imputation Using Deep Denoising Autoencoders [15], DL-GSA [7], and Swarm Intelligence-Deep Neural Network [27] can be powerful imputers. However, it is worth noting that deep learning methods usually require more data than statistical imputation approaches. Some other popularly used missing data imputation methods are multiple imputation by chained equation (MICE) [5], K-nearest Neighbors imputation (KNNI) [28], and mean imputation [28].

In addition, for the purpose of data imputation and data type, for classification, the impact of imputation techniques on different classifiers may vary. Specifically, [28] compares the performance of logistic regression with regularization, k-nearest neighbours (kNN), random forest, classification tree, and xgboost classifiers [28] on datasets with missing entries. They use different imputation methods (mean imputation/ MICE imputation [5]/ missForest [10]/ random imputation/ softImpute

[20]/ hot deck imputation, kNN imputation) and compare the performance. According to the paper, mean imputation seems to outperform other counterparts for logistic regression with regularization and kNN, random imputation wins for random forest, missForest seems to be the best imputer for classification tree, and hot deck imputation is the best for xgboost.

With the rapid growth of data size [11, 12], it is necessary to speed up the available imputation methods because many current approaches remain too slow for big datasets, as pointed out in an example in Section 1. This is where a popular dimension reduction method like PCA can come to use. PCA projects the original higher-dimensional dataset into a representation of lower dimensionality by extracting and retaining important information from the data and expressing this new information based on a set of orthogonal vectors known as principal components. Its goal is to find linear transformations of the original data that retain the maximal amount of variance. Note that there are some works on PCA under data missingness. For example, [13] considers the problem of finding principal components as an optimization problem of an objective function and proposes iterative solutions to it. On the other hand, [29] proposes a multiple imputation method for the estimates of the parameters (components and axes) of PCA to take into account the variability due to missing values. However, our work is different from these works in the sense that they target the problem of how to perform PCA for a dataset with missing data. Meanwhile, our frameworks utilize PCA to speed up the imputation processor to reduce the ratio between the number of features and the sample size.

# 3 Preliminaries

Let $\mathbf{X} = [x_{ij}]$ where $i = 1, ..., n; j = 1, ..., p$ be a input data matrix of $n$ samples, $p$ features. In addition, assume that the features are centered and scaled. We review two popular formulations of PCA, which we refer to as PCA formulation 1 (**PCA-form1**) and PCA formulation 2 (**PCA-form2**).

## 3.1 PCA based on covariance matrix (PCA-form1)

Let $\mathbf{\Sigma}$ be the covariance matrix of $\mathbf{X}$. Next, let $(\lambda_1, \mathbf{v}_1), ..., (\lambda_p, \mathbf{v}_p)$ be the sorted eigenvalue-eigenvector pairs of $\mathbf{\Sigma}$ such that $\lambda_1 \geq \lambda_2 \geq ... \geq \lambda_p \geq 0$. Suppose that we choose the first $r$ pairs for dimension reduction. Then the amount of variance explained by these $r$ pairs is

$$\frac{\lambda_1 + \lambda_2 + ... + \lambda_r}{\lambda_1 + \lambda_2 + ... + \lambda_p} \tag{1}$$

In addition, let $\mathbf{V} = [\mathbf{v}_1, \mathbf{v}_2, ..., \mathbf{v}_r]$. Then the dimension reduced version of $\mathbf{X}$ is $\mathbf{XV}$.

## 3.2 PCA based on the input matrix X (PCA-form2)

The solution of PCA can also be produced based on the singular value decomposition of $\mathbf{X}$ [30]:

$$\mathbf{X} = \mathbf{UDW}^T \tag{2}$$

where $\mathbf{U}$ is an $n \times p$ orthogonal matrix, $\mathbf{W}$ is a $p \times p$ orthogonal matrix, and $\mathbf{D}$ is a $p \times p$ diagonal matrix whose diagonal elements are $d_1 \geq d_2 \geq ... \geq d_p \geq 0$. Suppose that $r$ eigenvalues are used, then the projection matrix is $\mathbf{V} = \mathbf{W}_r \mathbf{W}_r^T$ where $\mathbf{W}_r$ consists of the first $r$ columns of $\mathbf{W}$. Then the dimension reduced version of $\mathbf{X}$ is also $\mathbf{XV}$.

# 4 PCA Imputation (PCAI)

In this section, we detail our PCAI framework, a PCA based framework that is capable of significantly improving the imputation speed of an imputer for high dimensional data, alleviating the memory issue for many approaches.

To start with some notations, let $pca(A)$ be a function of a data matrix $A$. The function returns $(\mathcal{R}_A, V)$ where $\mathcal{R}_A$ is the PCA-reduced version of $A$, and $V$ is the projection matrix where the $i^{th}$ column of $V$ is the eigenvector corresponding to the $i^{th}$ largest eigenvalue. In addition, denote by $\mathcal{A} \cup \mathcal{B}$ the columnwise concatenation of two data partition $\mathcal{A}$ and $\mathcal{B}$ of relevant sizes. Next, suppose that we have a dataset $\mathcal{D} = \mathcal{F} \cup \mathcal{M}$, where $\mathcal{F}$ consists of data from fully observed features and $\mathcal{M}$ consists of data from features with missing values.

The framework is as depicted in Algorithm 1. We first conduct dimension reduction on the fully observed partition $\mathcal{F}$, which produces a reduced version $\mathcal{R}$ of $\mathcal{F}$. Then, the imputation of $\mathcal{M}$ is done on the set $\mathcal{R} \cup \mathcal{M}$ instead of $\mathcal{D} = \mathcal{F} \cup \mathcal{M}$ as how imputations are usually done (i.e., impute directly on the original missing data). In conducting dimension reduction, we expect to reduce the dimension of the fully observed partition so that the imputation of $\mathcal{M}$ can be faster.

---

**Algorithm 1** PCAI framework

---

**Require:**
   - $\mathcal{D} = \mathcal{F} \cup \mathcal{M}$ where $\mathcal{F}$ is the fully observed partition and $\mathcal{M}$ is the partition with missing values
   - Imputer $I$
   - PCA algorithm $pca$
**Procedure:**
   $(\mathcal{R}, V) \leftarrow pca(\mathcal{F})$
   $\mathcal{M}' \leftarrow$ the imputed version of $\mathcal{M}$ based on $\mathcal{R} \cup \mathcal{M}$
   **return** Imputed version $\mathcal{M}'$ of $\mathcal{M}$

---

For the choice of the PCA formulation, note that if the number of samples is larger than the number of features in $\mathcal{F}$, then the size of the covariance matrix is smaller than the size of $\mathcal{F}$. Therefore, one may expect using the formulation of PCA based on the covariance matrix, as in Section 3.1, to be faster. Meanwhile, if the number of features in $\mathcal{F}$ is larger than the sample size, then the covariance matrix of $\mathcal{F}$ is larger than $\mathcal{F}$. Therefore, in such a case, it is better to use the PCA formulation based on the data itself, i.e., formulation as in Section 3.2.

One may reckon that using $\mathcal{R} \cup \mathcal{M}$ instead of $\mathcal{F} \cup \mathcal{M}$ may lead to loss of information due to dimension reduction and therefore lower the quality of imputation. However, as will be illustrated in the experiments, the differences between the mean squared error of the imputed version versus the ground truth for these approaches are only slightly different, and many times, PCAI seems to be slightly better. This is possibly because PCA retains the important information from the data while removing some noise, and therefore helps improving the imputation quality. However, PCAI also has some shortcomings. For problems where the sample size $n$ is smaller than the number of features in the fully observed block $q$, if PCA-form1 is used, the covariance matrix has the size of $q \times q$, which is bigger than the size $n \times q$ of the fully observed partition $\mathcal{F}$. This may make the PCA dimension reduction process become computationally expensive, rendering PCAI to be slower than imputing directly on the original missing data. This issue will be illustrated in the experiment section.

## 5 PCAI for classification (PIC)

In this section, we discuss a straightforward application of PCAI in classification, with a slight modification for classification problems where it is desirable to conduct a dimension reduction before training a model, such as when the number of features is much larger than the sample size.

Since PCAI conducts PCA on the fully observed partition $\mathcal{F}$, it reduces the dimensions for a portion of the data. Therefore, rather than imputing values using the PCAI framework and then conducting a dimension reduction step on $\mathcal{F} \cup \mathcal{M}'$, one can perform dimension reduction on $\mathcal{M}'$ to get $\mathcal{R}'$, a PCA-reduced version of $\mathcal{M}'$. Then, one can use $\mathcal{F} \cup \mathcal{R}'$ as reduced dimension data. As will be shown in the experiments, this speeds up the imputation and classification process significantly. This is the basic idea of our *Principle component Imputation for Classification (PIC)* framework.

PIC operates as shown in Algorithm 2. The procedure starts by performing PCA on the training fully observed partition $\mathcal{F}_{train}$, which gives the reduced version $\mathcal{R}_{train}$ of $\mathcal{F}_{train}$ and a projection matrix $V$. Next, we project $\mathcal{F}_{test}$ on $V$ to get the reduced version $\mathcal{R}_{test}$ of $\mathcal{F}_{test}$. Then, we impute $\mathcal{M}_{train}$ on $\mathcal{R}_{train} \cup \mathcal{M}_{train}$ to get the imputed version $\mathcal{M}'_{train}$. Next, we impute $\mathcal{M}_{test}$ on $\mathcal{R}_{test} \cup \mathcal{M}_{test}$ to get the imputed version $\mathcal{M}'_{test}$. After that, if $reduce_{miss}$ is set to true, we perform dimension reduction on $\mathcal{M}'_{train}, \mathcal{M}'_{test}$. Then, we train the classifier on $\mathcal{R}_{train} \cup \mathcal{R}'_{train}$, i.e., the union of the reduced version of $\mathcal{F}_{train}$ and the reduced version of $\mathcal{M}_{train}$. For prediction of a vector $\mathbf{x} \in \mathcal{D}$, we can decompose $\mathbf{x}$ into $\mathbf{x} = (\mathbf{x}_{\mathcal{F}}, \mathbf{x}_{\mathcal{M}})$. After that, we can project $\mathbf{x}_{\mathcal{F}}$ on $V$ to get a projection $\mathbf{r}$. Similarly, we can project $\mathbf{x}_{\mathcal{M}}$ on $V$ to a get projection $\mathbf{r}'$. Finally, we can predict the label of $\mathbf{x}$ using the classifier $C$ with input $(\mathbf{r}, \mathbf{r}')$.

175 Note that $reduce_{miss}$ is an option. When the number of features in the missing partition $\mathcal{M}$ is
176 large, one may be interested in reducing the dimension of $\mathcal{M}'$, and therefore, set $reduce_{miss}$ to *True*.
177 However, when the number of features in the missing partition is small, one may want to keep it to
178 *False*. Also, since PIC is a straightforward application of PCAI for classification, the choice of PCA
179 formulation should be used is similar to PCAI, which is analyzed in the previous section.

---

**Algorithm 2** PIC framework

---

**Require:**
   - $\mathcal{D} = \mathcal{F} \cup \mathcal{M}$ where $\mathcal{F}$ is the fully observed partition and $\mathcal{M}$ is the partition with missing values
   - $reduce_{miss} = True/False$: if *True*, perform dimension reduction on the imputed partitions; if
   *False*, do not perform dimension reduction on the imputed partitions
   - $\mathcal{F}_{train}, \mathcal{F}_{test}$: the training and testing data of the fully observed partition $\mathcal{F}$, respectively
   - $\mathcal{M}_{train}, \mathcal{M}_{test}$: the training and testing data of the partition that has missing data $\mathcal{M}$, respectively

   - Imputer $I$, classifier $C$, PCA algorithm $pca$
**Procedure:**
   $(\mathcal{R}_{train}, V) \leftarrow pca(\mathcal{F}_{train})$
   $R_{test} \leftarrow \mathcal{F}_{test}V$
   $\mathcal{M}'_{train} \leftarrow$ imputed version of $\mathcal{M}_{train}$ based on $\mathcal{R}_{train} \cup \mathcal{M}_{train}$
   $\mathcal{M}'_{test} \leftarrow$ imputed version of $\mathcal{M}_{test}$ based on $\mathcal{R}_{test} \cup \mathcal{M}_{test}$
   **if reduce$_{miss}$ then**
      $(\mathcal{R}'_{train}, W) \leftarrow pca(\mathcal{M}'_{train})$
      $\mathcal{R}'_{test} \leftarrow \mathcal{M}'_{test}V$
      Train the classifier $C$ based on $\mathcal{R}_{train} \cup \mathcal{R}'_{train}$
      Classify based on $\mathcal{R}_{test} \cup \mathcal{R}'_{test}$,
   **else**
      Train the classifier based on $\mathcal{R}_{train} \cup \mathcal{M}'_{train}$
      Classify based on $\mathcal{R}_{test} \cup \mathcal{M}'_{test}$
   **end if**
   **return** trained classifier $C$

---

180 ## 6   Relation to previous works

181 Various works have been done on PCA that are related to missing data, which mostly can be
182 categorized into missing values imputation using PCA, or dimension reduction using PCA under
183 missing values. Some typical works that make use of PCA for missing values imputation are
184 probabilistic PCA for missing flow volume data imputation [31]; chunk-wise iterative PCA for
185 data imputation on datasets with many samples[32]; [14] proposes a fast algorithm for PCA under
186 missing data that help in case of sparse, high dimensional data; [33] analyze maximum likelihood
187 PCA (MLPCA) on maximum likelihood missing data imputation; and [34] proposed an imputation
188 approach based on PCA and factorial analysis for mixed data.

189 Next, PCA under missing values was first studied in [35], where only one component and one
190 imputation iteration are used. After that, [36] proposes a method based on MLPCA, where the
191 method assigns large variance to missing values prior to implementing the method, which aim to
192 guide the algorithm to fit a PCA model disregarding those points. Also, [37] introduce EM algorithm
193 for building a PCA model that can deal with missing data. More recently, [38] proposes new
194 techniques for building a PCA model with missing data: known data regression (KDR), projection to
195 the model plane, KDR with principal component regression.In addition, [39] studies estimation and
196 imputation in Probabilistic PCA when the data is missing not at random.

197 Different from the previous approaches, PCAI is a framework to speed up the imputation process,
198 which can be used with various imputation methods, including the aforementioned PCA imputation
199 algorithms and the state-of-the-art imputation algorithms such as softImpute [20], MissForest [10],
200 GAIN [6]. In addition, note that since PCAI and PIC conduct dimension reduction on the fully
201 observed partition $\mathcal{F}$, and not the missing portion $\mathcal{M}$ if $reduce_{miss} = False$, they can handle missing
202 data even if categorical features presents in the missing portion $\mathcal{M}$, when being used with imputers
203 that's capable of handling categorical/mixed data (MissForest [10], SICE [24], FEMI [25], etc.). In

Table 1: Description of datasets used in our experiments

| Dataset | # Classes | # Features | # Samples |
|---|---|---|---|
| Parkinson [42] | 2 | 754 | 756 |
| Fashion MNIST [9] | 10 | 784 | 70000 |
| Gene [43] | 5 | 20531 | 801 |

addition, even if there exists categorical and continuous features in $\mathcal{M}$; or $reduce_{miss} = True$ and there exists categorical and continuous features in $\mathcal{M}$, one can easily adjust the algorithm to conduct PCA on continuous features only. The previously mentioned PCA based approaches are, however, can only be used for continuous data, because PCA requires the data to be continuous.

# 7 Experiments

## 7.1 General experiment settings

We compare the speed (seconds) and MSE of PCAI with **direct imputation (DI)**, i.e., use an imputation algorithm directly on the dataset. The imputation approaches used for comparison: softImpute [20, 40], MissForest [10] [1] and Multiple Imputation by Chained Equation (MICE) [5, 41], kNN Imputation (KNNI), GAIN [6] are implemented with default configurations. The codes will be available upon the acceptance of the paper. For PIC, we compare the five fold cross-validation (CV) score (accuracy, speed) of PIC when dimension reduction is applied on the imputed missing part (**PIC-reduce**), when dimension reduction is not applied on the imputed missing part (**PIC**), and when PCA is applied to the imputed version on the full missing data (**DI-reduce**), and when no dimension reduction is applied to imputed data after direct imputation (**DI**). Here, the default PCA formulation is PCA-form1, unless specified otherwise. For all PCA computation, the number of eigenvectors is chosen so that the minimum amount of variance explained is 95%.

Details of the datasets used in the experiments are available in Table 1. All experiments are run on an AMD Ryzen 7 3700X CPU with 8 Cores, 16 processing threads, 3.6GHz, and 16GB RAM.We terminate an experiment if no result is produced after 6,500 seconds of running or if there arises a memory allocating issue, and we denote this as **NA** in the result tables.

## 7.2 Performance of PCAI and PIC when the missing values in $\mathcal{M}$ are randomly simulated

Table 2: (MSE, speed) for PCAI and direct imputation (DI) on the Parkinson dataset with $q = 700$.

| Imputer | Strategy | missing rate | | |
|---|---|---|---|---|
| | | 20% | 40% | 60% |
| softImpute | PCAI | (0.073, 0.860) | (0.185, 0.774) | (0.305, 0.875) |
| | DI | (0.072, 4.097) | (0.188, 4.043) | (0.308, 4.467) |
| MICE | PCAI | (0.091, 139.811) | (0.186, 85.241) | (0.369, 109.815) |
| | DI | NA | NA | NA |
| GAIN | PCAI | (0.254, 45.046) | (0.538, 43.938) | (0.779, 43.956) |
| | DI | (0.608, 69.839) | (1.097, 70.548) | (1.369, 70.293) |
| missForest | PCAI | (0.064, 188.324) | (0.163, 178.849) | (0.292, 138.085) |
| | DI | (0.058, 905.002) | (0.160, 692.150) | (0.258, 449.415) |
| KNNI | PCAI | (0.127, 0.355) | (0.299, 0.398) | (0.466, 0.416) |
| | DI | (0.113, 0.310) | (0.274, 0.337) | (0.426, 0.372) |

Note that any datasets can be rearranged so that the first $q$ features are not missing and the remaining ones are missing. Therefore, without loss of generality, we assume that the first $q$ features of each

---

[1] https://pypi.org/project/missingpy/

Table 3: (MSE, speed) for PCAI and DI on the Fashion MNIST dataset with $q = 700$. MissForest results all are NA, and therefore are removed from the tables.

| Imputer | Strategy | missing rate | | |
| --- | --- | --- | --- | --- |
| | | 20% | 40% | 60% |
| softImpute | PCAI | (0.032, 22.408) | (0.066, 22.797) | (0.109, 25.603) |
| | DI | (0.032, 67.627) | (0.064, 69.349) | (0.107, 77.233) |
| MICE | PCAI | (0.027, 2218.864) | (0.055, 1374.558) | (0.095, 1641.962) |
| | DI | NA | NA | NA |
| GAIN | PCAI | (0.053, 65.730) | (0.091, 68.752) | (0.137, 69.743) |
| | DI | (0.041, 97.898) | (0.079, 99.049) | (0.125, 96.317) |
| KNNI | PCAI | (0.055, 1607.850) | (0.115, 2033.153) | (0.180, 2272.370) |
| | DI | (0.049, 3042.752) | (0.102, 3659.300) | (0.161, 3959.832) |

228 dataset are not missing, and the remaining ones contain missing value(s). Then, we simulated missing
229 data randomly on the missing partition $\mathcal{M}$ with missing rates 20%, 40%, and 60%. Here, a missing
230 rate of 20% means that 20% of the entries in the missing partition $\mathcal{M}$ are missing. The results for
231 such experiments are reported in Tables 2, 3, 4. Due to space limit, the results related to PIC on
Fashion MNIST are reported in the Appendix.

Table 4: Five fold CV results (accuracy, speed) of SVM on Parkinson with $q = 700$.

| Imputer | Strategy | missing rate | | |
| --- | --- | --- | --- | --- |
| | | 20% | 40% | 60% |
| softImpute | PIC-reduce | (0.862, 1.026) | (0.862, 1.137) | (0.862, 1.161) |
| | PIC | (0.858, 1.008) | (0.858, 1.079) | (0.859, 1.112) |
| | DI-reduce | (0.861, 4.116) | (0.862, 4.424) | (0.861, 4.718) |
| | DI | (0.858, 3.775) | (0.858, 3.912) | (0.855, 4.248) |
| MICE | PIC-reduce | (0.859, 204.605) | (0.861, 256.340) | (0.861, 240.211) |
| | PIC | (0.858, 524.739) | (0.859, 694.667) | (0.859, 925.426) |
| | DI-reduce | NA | NA | NA |
| | DI | NA | NA | NA |
| GAIN | PIC-reduce | (0.857, 91.086) | (0.852, 102.861) | (0.848, 122.349) |
| | PIC | (0.851, 89.984) | (0.853, 104.773) | (0.853, 123.233) |
| | DI-reduce | (0.855, 130.349) | (0.851, 149.864) | (0.851, 181.135) |
| | DI | (0.846, 129.702) | (0.849, 152.031) | (0.852, 183.67) |
| missForest | PIC-reduce | (0.859, 204.850) | (0.861, 276.537) | (0.858, 153.783) |
| | PIC | (0.858, 202.939) | (0.861, 277.067) | (0.858, 153.463) |
| | DI-reduce | (0.861, 656.948) | (0.862, 729.872) | (0.861, 472.230) |
| | DI | (0.858, 655.750) | (0.861, 730.013) | (0.858, 472.388) |
| KNNI | PIC-reduce | (0.858, 0.533) | (0.861, 0.462) | (0.862, 0.625) |
| | PIC | (0.858, 0.513) | (0.861, 0.462) | (0.862, 0.607) |
| | DI-reduce | (0.862, 0.696) | (0.862, 0.642) | (0.859, 0.803) |
| | DI | (0.859, 0.438) | (0.859, 0.45) | (0.858, 0.552) |

232
233 From the tables, it is clear that the proposed frameworks reduce the imputation time significantly
234 while maintaining competitive MSE/classification accuracy compared to DI, in most of the cases.
235 For example, at the missing rate 20% on the Parkinson dataset (Table 4), when using GAIN for
236 imputation, the running time of PIC-reduce(91.086s) is much lower compared to DI-reduce (130.349),
237 the running time of PIC (89.984s) is also much lower compared to DI (129.702). Another example
238 can be seen from Table 2, for the Parkinson dataset, at 20% missing rate, when PCAI is applied to
239 missForest, the running time reduces to 188.324s, which is almost $1/5$ of the DI (905.002s). Next,

on Fashion MNIST (Table 3), it is worth noticing that for MICE, DI cannot gives the results due to memory issue but PCAI can alleviate this issue and deliver the results.

For KNNI, the running time for KNNI between the PCAI approach and direct imputation for Parkinson (Table 2) is not much different. However, for the Fashion MNIST dataset, KNNI using the PCAI framework obviously deliver a competitive result in a significantly shorter time. Specifically, KNNI at a missing rate of 20% on Fashion MNIST gives a result after only 1607.850 seconds, while DI takes up to 3,042.752 seconds. This is because Fashion MNIST (70000 samples) has much more samples than Parkinson (756 samples), and KNN need to do a lot of pairwise comparison. Therefore, PCAI and PIC would be extremely helpful for KNNI when the sample size and the number of fully observed features is large. Note that it does not require the number of features with missing data to be large or small.

From Table 2, we can see that PCAI generates a lot of improvements in MSE for GAIN, in addition to improvements in speed. This is possibly because PCA reduces the number of features while the sample size remains the same, making such a deep learning approach more applicable to the newly reduced data.

## 7.3 Performance on nonrandomly missing data

In many fields, the data are missing in a monotone pattern rather than random [44]. Therefore, we generate one-step monotone missing data on Fashion MNIST by first, randomly choose 20%, 40%, 60% of the samples. Then, we make them become missing by deleting the lower right corner by deleting the intersection between the last 8 rows and the last 13 columns of each image array. The results are reported in Table 5. From the table, we can see that PIC-reduce is a great improvement in speed compared to DI-reduce, and PIC is a significant improvement in speed compared to DI. This illustrates that PIC can work effectively even for non-randomly missing data.

Table 5: Five fold CV results (accuracy, speed) of SVM on monotone data generated on Fashion MNIST.

| Imputer | Strategy | missing rate | | |
| | | 20% | 40% | 60% |
| --- | --- | --- | --- | --- |
| softImpute | PIC-reduce | (0.889, 409.676) | (0.889, 421.671) | (0.889, 369.49) |
| | PIC | (0.889, 507.626) | (0.89, 543.309) | (0.889, 480.452) |
| | DI-reduce | (0.89, 439.268) | (0.89, 528.892) | (0.889, 395.646) |
| | DI | (0.891, 738.494) | (0.891, 872.616) | (0.89, 646.781) |
| GAIN | PIC-reduce | (0.886, 462.478) | (0.883, 429.173) | (0.882, 449.786) |
| | PIC | (0.886, 493.399) | (0.882, 484.232) | (0.881, 496.066) |
| | DI-reduce | (0.891, 543.803) | (0.89, 431.947) | (0.89, 454.981) |
| | DI | (0.892, 902.049) | (0.891, 754.794) | (0.891, 780.686) |

## 7.4 PIC under different PCA formulations and number of missing features

The missing data in these experiments are generated at random as in Section 7.2 and the five fold cross validation results of SVM on the Gene dataset with $q = 15000, 20000$, are shown in Table 6 and Table 7. From these tables, one can see clearly that for datasets where the number of features are significantly higher than the number of samples such as Gene, PCA-form2, which is based on the input data ($\mathcal{F}$ specifically) gives much faster computations compared to PCA-form1, and also is faster than direct imputation-classification without PCA. In addition, when PCA-form1 is used, even though PIC and PIC-reduce are faster than PCA on directly imputed data (DI-reduce), they are still much slower than direct imputation - classification without PCA.

Interestingly, the accuracy PIC and PIC-reduce are almost identical to PCA on directly imputed data, and are higher than direct imputation - classification without PCA. Next, note that the main idea of the proposed methods is to reduce the dimension of the $\mathcal{F}$ to speed up the imputation. Therefore, we have made no assumption about the number of features in the missing portion $\mathcal{M}$. In Table 6 and Table 7, $q = 15000, 20000$, which means 5,531 and 531 missing features in $\mathcal{M}$, respectively. This implies PIC can handle datasets where $\mathcal{M}$ has many features.

Table 6: Five fold CV results (accuracy, speed) of SVM for softImpute based strategies on the Gene dataset when $q = 15000$.

| | | missing rate | | |
|---|---|---|---|---|
| | Strategy | 20% | 40% | 60% |
| PCA-form1 | PIC-reduce | (0.994, 2250.451) | (0.992, 2412.082) | (0.992, 2415.434) |
| | PIC | (0.992, 2429.114) | (0.992, 2276.354) | (0.992, 2284.414) |
| | DI-reduce | (0.994, 5018.368) | (0.994, 4529.766) | (0.994, 3785.947) |
| PCA-form2 | PIC-reduce | (0.995, 69.444) | (0.992, 76.393) | (0.992, 85.2) |
| | PIC | (0.992, 61.451) | (0.992, 68.571) | (0.992, 77.528) |
| | DI-reduce | (0.995, 80.823) | (0.992, 92.265) | (0.994,100.751) |
| No PCA | DI | (0.985, 71.884) | (0.985, 74.812) | (0.985, 92.309) |

Table 7: Five fold CV results (accuracy, speed) of SVM for softImpute based strategies on the Gene dataset when $q = 20000$.

| | | missing rate | | |
|---|---|---|---|---|
| | Strategy | 20% | 40% | 60% |
| PCA-form1 | PIC-reduce | (0.994, 2578.910) | (0.994, 4001.717) | (0.994, 3848.950) |
| | PIC | (0.994, 2583.717) | (0.994, 4144.157) | (0.994, 4057.188) |
| | DI-reduce | (0.995, 2891.994) | (0.994, 4476.563) | (0.995, 4332.869) |
| PCA-form2 | PIC-reduce | (0.995, 67.753) | (0.992, 73.884) | (0.995, 81.27) |
| | PIC | (0.995, 59.815) | (0.995, 66.096) | (0.995, 73.079) |
| | DI-reduce | (0.995, 81.07) | (0.995, 82.407) | (0.995, 91.638) |
| No PCA | DI | (0.985, 74.06) | (0.985, 71.6) | (0.985, 84.963) |

# 8 Conclusion and Remarks

We have presented two novel frameworks for datasets where many continuous features are fully observed, PCAI and PIC, that can speed up imputation algorithms significantly while having competitive accuracy MSE/accuracy compared to direct imputation and alleviate the memory issue for some imputation approaches such as MICE, kNN. In addition, the frameworks can be used even when some or all of the missing features are categorical or when the number of missing features is large. Note that when the sample size is significantly larger than the number of fully observed features, PCA-form1 should be used since, in such a case, the covariance matrix is much smaller than $\mathcal{F}$, making it faster than PCA-form2. On the other hand, when the number of fully observed features is significantly larger than the sample size, PCA-form2 should be preferred, as the covariance matrix is bigger than $\mathcal{F}$ itself in such a case. A limitation of the proposed framework is that if there are not many fully observed continuous features, then due to the computational cost of PCA, the proposed frameworks may not lead to any improvement in speed.

Even though PIC is only introduced for classification, the same strategy can be applied to a regression problem. We would like to explore that in the future. Moreover, since various dimension reduction techniques such as sparse PCA [45], incremental PCA [46], truncated SVD [47] have been developed to suit different scenarios, it is worth investigating different dimension reduction techniques for PCAI and PIC. In addition, it would be interesting to explore if applying a PCA variant to the missing partition $\mathcal{M}$ would result in even a more efficient method for datasets with continuous features in the missing partition.

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
