# Supplementary materials for "Principal Components Analysis based frameworks for efficient missing data imputation algorithms"

1 In the following, we present the supplementary materials for the paper *Principal Components Analysis*
2 *based frameworks for efficient missing data imputation algorithms*.

3 ## A   PIC on the Fashion MNIST dataset

Table 1: 5 - fold cross validation results (accuracy, speed) of SVM for different imputation-classification strategies on the Fashion MNIST dataset with $q = 700$, when the missing values in $\mathcal{M}$ are simulated at random given rates.

| Imputer | Strategy | missing rate | | |
|---|---|---|---|---|
| | | 20% | 40% | 60% |
| softImpute | PIC-reduce | (0.891, 285.707) | (0.890, 279.263) | (0.889, 353.506) |
| | PIC | (0.891, 593.652) | (0.891, 598.726) | (0.890, 609.252) |
| | DI-reduce | (0.892, 467.406) | (0.891, 458.625) | (0.891, 486.162) |
| | DI | (0.892, 710.019) | (0.892, 644.585) | (0.891, 630.044) |
| MICE | PIC-reduce | (0.892, 2379.869) | (0.892, 1884.385) | (0.891, 2851.581) |
| | PIC | (0.892, 2416.485) | (0.891, 1907.529) | (0.892, 2897.839) |
| | DI-reduce | NA | NA | NA |
| | DI | NA | NA | NA |
| GAIN | PIC-reduce | (0.892, 539.720) | (0.891, 580.754) | (0.891, 660.633) |
| | PIC | (0.891, 583.016) | (0.892, 608.384) | (0.891, 703.145) |
| | DI-reduce | (0.891, 1320.018) | (0.891, 1351.854) | (0.890, 1573.656) |
| | DI | (0.891, 846.002) | (0.892, 787.246) | (0.891, 863.685) |
| missForest | PIC-reduce | NA | NA | NA |
| | PIC | NA | NA | NA |
| | DI-reduce | NA | NA | NA |
| KNNI | PIC-reduce | (0.891, 3040.393) | (0.891, 4235.292) | (0.89, 6059.608) |
| | PIC | (0.891, 3110.041) | (0.891, 4279.305) | (0.889, 6088.542) |
| | DI-reduce | NA | NA | NA |
| | DI | NA | NA | NA |

## B  PIC on Fashion MNIST under monotone missing data

Table 2 shows the full results for PIC on Fashion MNIST under monotone missing data, which corresponds to the experiment settings in *Section 7.3: Performance on nonrandomly missing data* in the paper.

Interestingly, in this case, for KNNI, at 20% missing rates, DI-reduce is faster than PIC-reduce, and DI is faster than DI. However, as the missing rates increase to 40% or 60%, PIC-reduce is significantly faster than PIC, and PIC is also significantly faster than DI. Also, note that in Table 1, when KNNI is used, DI and DI-reduce gives NA (due to memory issue). This shows that how the data is missing affects the performance of KNNI, and that PCAI and PIC have the ability to alleviate the memory requirements of KNNI.

Table 2: Five fold CV results (accuracy, speed) of SVM on monotone data generated on Fashion MNIST. MICE and MissForest are excluded from the table because all the corresponding results are NA.

| Imputer | Strategy | missing rate | | |
| --- | --- | --- | --- | --- |
| | | 20% | 40% | 60% |
| softImpute | PIC-reduce | (0.889, 409.676) | (0.889, 421.671) | (0.889, 369.49) |
| | PIC | (0.889, 507.626) | (0.89, 543.309) | (0.889, 480.452) |
| | DI-reduce | (0.89, 439.268) | (0.89, 528.892) | (0.889, 395.646) |
| | DI | (0.891, 738.494) | (0.891, 872.616) | (0.89, 646.781) |
| GAIN | PIC-reduce | (0.886, 462.478) | (0.883, 429.173) | (0.882, 449.786) |
| | PIC | (0.886, 493.399) | (0.882, 484.232) | (0.881, 496.066) |
| | DI-reduce | (0.891, 543.803) | (0.89, 431.947) | (0.89, 454.981) |
| | DI | (0.892, 902.049) | (0.891, 754.794) | (0.891, 780.686) |
| KNNI | PIC-reduce | (0.89, 2220.308) | (0.89, 2344.1) | (0.889, 2800.795) |
| | PIC | (0.89, 2278.491) | (0.891, 2381.274) | (0.89, 2838.674) |
| | DI-reduce | (0.89, 1720.832) | (0.89, 2614.603) | (0.89, 3173.561) |
| | DI | (0.891, 1958.2) | (0.891, 2854.299) | (0.89, 3416.44) |

## C  PCAI on Fashion MNIST under monotone missing data

Table 3 shows the full results for PCAI on Fashion MNIST under monotone missing data, which corresponds to the experiment settings in *Section 7.3: Performance on nonrandomly missing data* in the paper.

Table 3: (MSE, speed) for PCAI and DI on the Fashion MNIST dataset under monotone data. MICE and MissForest are excluded from the table because all the corresponding results are NA.

| Imputer | Strategy | missing rate | | |
| --- | --- | --- | --- | --- |
| | | 20% | 40% | 60% |
| softImpute | PCAI | (0.052, 50.057) | (0.098, 61.523) | (0.014, 63.632) |
| | DI | (0.050, 170.488) | (0.095, 212.620) | (0.013, 212.620) |
| GAIN | PCAI | (0.34, 67.052) | (0.793, 68.788) | (0.96, 67.247) |
| | DI | (0.367, 108.869) | (0.898, 106.761) | (1.424, 109.571) |
| KNNI | PCAI | (0.054, 1210.02) | (0.099, 1929.294) | (0.139, 2349.668) |
| | DI | (0.046, 2199.876) | (0.085, 3507.741) | (0.120, 4181.563) |