# OpenReview forum: "Principle Components Analysis based frameworks for efficient missing data imputation algorithms"
_NeurIPS.cc/2022/Conference — NeurIPS 2022 Submitted_

### Official Review · Reviewer_wGwK · 2022-07-04

**Rating:** 5
**Confidence:** 3
**Soundness:** 2 fair
**Presentation:** 2 fair
**Contribution:** 2 fair

**Summary:**

The authors propose we propose a framework based on PCA to speed up the imputation process of many available imputation techniques. The proposed methods are named PCAI and PCA-PIC. Experiments are provided on various datasets.

**Questions:**

Can the authors explain why the traditional style of imputing on the original missing dataset gives NAs for all missing rates?

**Limitations:**

The authors addressee adequately the limitaions of  PCAI and PCA-PI.

**Strengths And Weaknesses:**

Weaknesses:
1) The authors only compared PCAI strategy with the traditional strategy.
2) "Principle Component Analysis" should be "Principal Component Analysis"

Strengths:
The proposed methods are named PCAI and PCA-PI are novel.

---

> ### Author Response · Authors · 2022-08-02
> **Response to Reviewer [wGwK]**
>
> Thank you very much for your valuable comments. We addressed your comments and suggestions as follows,
>
> 1.	“The authors only compared PCAI strategy with the traditional strategy.”
>
> Response: Even though PCAI is only compared with the traditional strategy, the experiments cover various imputation algorithms, including the state-of-the-art imputation techniques such as GAIN[1], softImpute [2], missForest [3].  We also changed the term “traditional strategy” into “direct imputation (DI)” in this revision for a clearer meaning of the term.
> In addition, in this revision, we also added many experiments and analyses to validate our approaches. We also have added a section of relation to previous works, including many recent ones, to clarify the differences between the proposed approaches and what the research community has done so far. All these factors validate the promising potential of our frameworks.
>
> References:
>
> [1] Yoon, Jinsung, James Jordon, and Mihaela Schaar. "Gain: Missing data imputation using generative adversarial nets." International conference on machine learning. PMLR, 2018.
>
> [2] Mazumder, Rahul, Trevor Hastie, and Robert Tibshirani. "Spectral regularization algorithms for learning large incomplete matrices." The Journal of Machine Learning Research 11 (2010): 2287-2322.
>
> [3] Stekhoven, Daniel J., and Peter Bühlmann. "MissForest—non-parametric missing value imputation for mixed-type data." Bioinformatics 28.1 (2012): 112-118.
>
> 2.	"Principle Component Analysis" should be "Principal Component Analysis"
>
> Response: We fixed that error throughout the paper.
>
> 3.	“Can the authors explain why the traditional style of imputing on the original missing dataset gives NAs for all missing rates?”
>
> Response: We terminate an experiment if no result is produced after 6500 seconds of running or if a memory allocating issue arises, and we denote this as NA in the result tables. This is clarified and explained more clearly in the revised version.

---

### Official Review · Reviewer_SAVj · 2022-07-06

**Rating:** 4
**Confidence:** 4
**Soundness:** 3 good
**Presentation:** 2 fair
**Contribution:** 3 good

**Summary:**

The authors present a PCA based framework for missing data imputation specially efficient for high dimensional data. The experiments indicate that the imputation presents low MSE and, when applied to classification tasks, results in similar or better accuracy values.

**Questions:**

Some details of the proposed algorithms and experiments can be improved:

- Although it is not explicit in the text, I have inferred that the so-called "partition with missing values" correspond to features with unknown values (columns of the dataset), instead of samples with unknown feature values (rows of the dataset). Am I correct?
- In Algorithm 1, how exactly "R U M" is handled? Column-wise concatenation?
- How exactly the missing data was generated? It is not clear if a 20% missing rate is related to 20% instances with a single random missing feature or if it means that 20% of the total features are missing.
- In the classification experiments, how many data points are used for training and testing? How many independent runs? How hyperparameter selection was performed?

I do not think it is necessary to separate the PCAI and PIC frameworks. It seems to me that PIC is the straightforward application of PCAI on data related to a classification task. Unless I am mistaken, I recommend unifying the presentation of both strategies.

I list additional comments below:

- I believe "Principal Components Analysis" is preferred over "Principle Components Analysis".
- Line 117: "We first conduct dimension reduction on the fully observed partition M..." -> 'F' is the fully observed partition.
- Line 151: The matrix 'W' has not been defined at this point.
- Tab. 4 should include the standard result obtained by the SVM classifier, without PCA.
- I recommend always including in the supervised experiments the simple baseline of simply removing the instances with any missing data.

**Limitations:**

The manuscript states sufficiently its limitations.

**Strengths And Weaknesses:**

The main motivation is relevant and promising. The related work section indicates a well explored field. Nevertheless, the authors should mention and include in the experimental section at least some of the many PCA based approaches to data imputation. See for instance [1,2,3,4] and the references within. Without those comparisons, it is difficult to state the originality of the work.

Since the proposed approach relies on classic PCA and can be coupled with any imputation strategy, it seems to be readily available for practitioners. This is a welcomed advantage of the proposal.

The writing is mostly sound and the manuscript is well organized. However, some steps of the proposed algorithms are difficult to follow due to the lack of details. I will list them in the next part.

References

[1] Qu, Li, et al. "PPCA-based missing data imputation for traffic flow volume: A systematical approach." IEEE Transactions on intelligent transportation systems 10.3 (2009): 512-522.

[2] Folch-Fortuny, Abel, Francisco Arteaga, and Alberto Ferrer. "PCA model building with missing data: New proposals and a comparative study." Chemometrics and Intelligent Laboratory Systems 146 (2015): 77-88.

[3] Folch‐Fortuny, Abel, Francisco Arteaga, and Alberto Ferrer. "Assessment of maximum likelihood PCA missing data imputation." Journal of Chemometrics 30.7 (2016): 386-393.

[4] D’Enza, Alfonso Iodice, Francesco Palumbo, and Angelos Markos. "Single Imputation Via Chunk-Wise PCA." Data Analysis and Rationality in a Complex World (2021): 75.

---

> ### Author Response · Authors · 2022-08-02
> **Response to Reviewer [SAVj]**
>
> Thank you very much for your valuable comments and suggestions. We addressed your comments as follows:
>
> 1.	The main motivation is relevant and promising. The related work section indicates a well explored field. Nevertheless, the authors should mention and include in the experimental section at least some of the many PCA based approaches to data imputation. See for instance [1,2,3,4] and the references within. Without those comparisons, it is difficult to state the originality of the work.
>
> Response: We added section 6: “Relation to previous works,” to the paper to clarify the difference between PCAI, PIC, and other related works, including the works that you have mentioned. In this section, we also pointed out several advantages of those frameworks that were not pointed out in the initial submission, such as PCAI and PIC can be applied to data where categorial features appear in the missing partition when being used with imputers that are capable of handling categorical/mixed data such as MissForest.  The previously mentioned PCA-based approaches, however, can only be used for continuous data because PCA is designed for continuous data.
>
> 2.	Although it is not explicit in the text, I have inferred that the so-called "partition with missing values" correspond to features with unknown values (columns of the dataset), instead of samples with unknown feature values (rows of the dataset). Am I correct?
>
> Response: Yes. That is true. We have added the details at the end of the second paragraph in Section 4 “PCA Imputation (PCAI).”
>
> 3.	In Algorithm 1, how exactly "R U M" is handled? Column-wise concatenation?
>
> Response: Yes, it is column-wise concatenation. We have added the details in the second paragraph in Section 4 “PCA Imputation (PCAI)”.
>
> 4.	How exactly the missing data was generated? It is not clear if a 20% missing rate is related to 20% instances with a single random missing feature or if it means that 20% of the total features are missing.
>
> Response: Here, a missing rate of 20% means that 20% of the entries in the missing partition $\mathcal{M}$ are missing. We added this detail to the second paragraph of section 7.1.
>
> 5.	In the classification experiments, how many data points are used for training and testing? How many independent runs? How hyperparameter selection was performed?
>
> Response: In the classification experiments, we used 5- fold cross validation and reported the mean of the accuracy and running time. For PCA, we selected the number of eigenvalues – eigenvectors so that the selected can explain at least 95% of the variance. The number of $q$ is also added to the caption of each table. We added this information to the revision.
>
> 6.	I do not think it is necessary to separate the PCAI and PIC frameworks. It seems to me that PIC is the straightforward application of PCAI on data related to a classification task. Unless I am mistaken, I recommend unifying the presentation of both strategies.
>
> Response: PIC is truly a straightforward application of PCAI for classification, with only some slight modifications. Therefore, we changed the title of Section 5 from “PCA Imputation-Classification (PIC)” to “PCAI for classification (PIC),” and added a paragraph to the beginning of the section to emphasize the connection. However, we kept them as two frameworks because PIC requires splitting training and testing sets. In addition, it is easier to present and analyze the experiment results if different notations are used.
>
> 7.	I believe "Principal Components Analysis" is preferred over "Principle Components Analysis".
>
> Response: We have fixed the error throughout the paper.
>
> 8.	Line 117: "We first conduct dimension reduction on the fully observed partition M..." -> 'F' is the fully observed partition.
>
> Response: We have fixed the error you mentioned.
>
> 9.	Line 151: The matrix 'W' has not been defined at this point.
>
> Response: It is supposed to be “V” instead of “W.” We fixed this in the revised version.
>
> 10.	Tab. 4 should include the standard result obtained by the SVM classifier, without PCA.
>
> Response: We have added such results to all the experiments related to classification.
>
> 11.	I recommend always including in the supervised experiments the simple baseline of simply removing the instances with any missing data.
>
> Response: Due to limited time, we are not able to add these experiments. However, we will include that for future work.

---

> > ### Comment · Reviewer_SAVj · 2022-08-09
> > **Thanks for the answers**
> >
> > I would like to thank the authors for answering my questions and fixing the issues that I have pointed out.
> >
> > The new section on related work is very welcomed, but I believe there is still room for comparing performance with simple baselines and pursuing alternative simpler methods for speed gains (as mentioned by reviewer BvMn).

---

> > > ### Author Response · Authors · 2022-08-09
> > > **Response to Reviewer [SAVj]**
> > >
> > > Thank you very much for your comments. We do believe that it would be interesting to explore such settings and to explore the use of SVD and LDA as alternatives to PCA, since LDA is a popular supervised dimension reduction technique that makes use of the information from the labels, and  SVD is another commonly used technique for sparse data. Therefore, we would like to study how they can be applied to speed up imputation deeply for a supervised learning problem/for problems with sparse data, in the future.

---

### Official Review · Reviewer_emZM · 2022-07-19

**Rating:** 4
**Confidence:** 4
**Soundness:** 2 fair
**Presentation:** 2 fair
**Contribution:** 2 fair

**Summary:**

The authors presented two novel frameworks, PCAI and PIC, that allow for more efficient handling of missing data. There are two use cases, fashion MNIST (imaging data) and Parkinson (voice recording) dataset.

**Questions:**

Testing and optimizing imputation is complex and unless one is taking sufficient steps, the results may be biased, and the study will have limited utility and generalizability.
A comparison of a fashion dataset (in this case imaging data) and a medical dataset (in this case, voice recording) cannot be only about the number of records and number of features. To impute one has to understand the data characteristics, beyond simple statistics. A Deeper discussion of these differences is important if such comparisons are made.

**Limitations:**

The main assumption is that introducing random missing is not acceptable unless the author can show that such data suffers from random missing. For instance, clinical data has missing that is not completely random. Please see the paper published in nature digital medicine on this topic. https://www.nature.com/articles/s41746-021-00518-0
Similarly, voice recordings and imaging data suffer likely from non at random missingness due to the nature of the data.

**Strengths And Weaknesses:**

Strength: easy to understand and implement.
Weakness: The main assumption is that introducing random missing is not acceptable without sufficient evidence and justification.

---

> ### Author Response · Authors · 2022-08-02
> **Response to Reviewer [emZM]**
>
> Thank you very much for your valuable comments. Random missing is not an assumption for PCAI and PIC to work. We meant to use these datasets as illustrations and benchmarks only. We agree that images and audio data usually do not suffer from random missingness. However, when there are bad connections in streaming services, then random missingness may also. In addition, we also added to this revision experiments on the Gene dataset as an example for tabular data.
>
> After going through the paper you mentioned, we realized that in many fields, monotone missing data is a common issue. Even though randomly missing data is also a common problem, as mentioned in [1], we should not limit ourselves to such a setting. Therefore, we added experiments on the Fashion MNIST dataset, where the right lower corner of the image is missing. This corresponds to a one-step monotone missing pattern. The results of this experiment show that our framework is not limited to randomly missing data.
>
> Reference:
>
> [1] Little, Roderick JA, and Donald B. Rubin. Statistical analysis with missing data. Vol. 793. John Wiley & Sons, 2019.

---

> > ### Author Response · Authors · 2022-08-09
> > **Response to Reviewer [emZM]**
> >
> > Once again, thank you very much for your valuable comments. We also have added the tables of all the results of PIC and PCAI for Fashion MNIST under monotone missing data in the Appendix.

---

### Official Review · Reviewer_BvMn · 2022-07-22

**Rating:** 2
**Confidence:** 4
**Soundness:** 2 fair
**Presentation:** 3 good
**Contribution:** 1 poor

**Summary:**

This paper proposes a framework based on principle components analysis (PCA) to speed up the missing data imputation. It divides the feature sets into two partitions -- the fully observed one and the one that contains missing values. The proposed method applies PCA to the fully observed partition to do dimensionality reduction, followed by the existing imputation methods. The authors further propose to apply PCA to the imputed data to speed up the downstream classification task.

**Questions:**

A suggestion for further improvement: the proposed method only applies PCA to the fully observed partition, yet in many important cases, either we have no prior information about which feature is fully observed, or the feature partition with missing value is very large. If the PCA variant to be proposed could directly apply to the partition with missing values, it would much more interesting. Of course, it has to come with theoretical and empirical validations.

**Limitations:**

Limitations are not discussed in this paper.

**Strengths And Weaknesses:**

Strengths:
- The paper is overall organized and readers can easily follow.

Weaknesses:
- The major weakness is that the methodological contribution is quite limited. Projecting data into lower dimensional spaces to speed up downstream tasks is not new. In fact, it has been widely regarded as the go-to tool to handle large datasets.

---

> ### Author Response · Authors · 2022-08-02
> **Response to Reviewer [Reviewer BvMn]**
>
> Thank you very much for your valuable comments. We addressed your comments as follows:
>
> 1.	The major weakness is that the methodological contribution is quite limited. Projecting data into lower dimensional spaces to speed up downstream tasks is not new. In fact, it has been widely regarded as the go-to tool to handle large datasets
>
> Response: Thank you very much. We agree that PCA is a commonly used tool for large datasets. Hence, various works on PCA under missing data [1, 2, 4] and many missing data imputation methods based on PCA [2, 3] have been developed.
> However, up to our knowledge, none of the research works so far has used PCA to boost the imputation speed for missing data imputation and study this matter as extensive as this work. Meanwhile, missing data for high dimensional data is an important problem because data nowadays tend toward high dimensions. Moreover, our method, as pointed out by the third reviewer, “relies on classic PCA and can be coupled with any imputation strategy, it seems to be readily available for practitioners. This is a welcomed advantage of the proposal.” To clarify the distinction and capabilities of our frameworks compared to previous works, we have added a section that details the relation of our works to those prior works, including the references below. Moreover, in this revision, we have rewritten the abstract, analyzed more aspects of the matter such as the choice of PCA formulations, usages when there is categorical/mixed data in the features with missing data, the number of features in the missing partition $\mathcal{M}$, and run more experiments to highlight the potentials of the proposed frameworks.
>
> References:
>
> [1] Roweis, Sam. "EM algorithms for PCA and SPCA." Advances in neural information processing systems 10 (1997).
>
> [2] Sportisse, Aude, Claire Boyer, and Julie Josse. "Estimation and imputation in probabilistic principal component analysis with missing not at random data." Advances in Neural Information Processing Systems 33 (2020): 7067-7077.
>
> [3] Qu, Li, et al. "PPCA-based missing data imputation for traffic flow volume: A systematical approach." IEEE Transactions on intelligent transportation systems 10.3 (2009): 512-522.
>
> [4] Ilin, Alexander, and Tapani Raiko. "Practical approaches to principal component analysis in the presence of missing values." The Journal of Machine Learning Research 11 (2010): 1957-2000.
>
> 2.	A suggestion for further improvement: the proposed method only applies PCA to the fully observed partition, yet in many important cases, either we have no prior information about which feature is fully observed, or the feature partition with missing value is very large. If the PCA variant to be proposed could directly apply to the partition with missing values, it would much more interesting. Of course, it has to come with theoretical and empirical validations
>
> Response: Thank you for the suggestion. We agree that a PCA-based method that can be applied directly to the features with missing data would be very interesting, and we would like to study that extensively in future work. The method developed in the paper does not address this issue and is applicable if we know which features are fully observed or not. Since the main idea is to reduce the size of the fully observed partition $\mathcal{F}$, it is useful when the number of fully observed features is large. In this revision, we added experiments on the Gene dataset, with up to 5531 features with missing values to illustrate that PIC can work effectively when the number of features with missing values is large.
>
> However, as pointed out above, our approach still represents a significant contribution compared to existing imputation methods, especially with the focus on improving computational speed. Even though PCA is both a popular and common standard, using it in the way that we suggest in the paper is far from standard. Thus, the novelty in the paper is not in using PCA per se but in how we use it. The significance of the suggested method is further backed up by the experimental results.
>
> We also would like to highlight beneficial points of our frameworks that we have added to our revision (Section 6: Relation to previous works): PCAI and PIC are applicable even when the missing portion has categorical features, if being used with imputers that are capable of handling categorical/mixed data such as missForest, SICE, FEMI. Moreover, even if there exist categorical and continuous features in $\mathcal{M}$; or $reduce_{miss} = True$ and there exists categorical and continuous features in $\mathcal{M}$, one can adjust the algorithm to conduct PCA on continuous features only. Other works on PCA that are related to missing data are, however, up to our knowledge, cannot be used directly when there are categorical features in the data, as PCA is designed for continuous data.
>
> 3.	Limitations are not discussed in this paper.
>
> Response: We have added the limitations in the conclusion section.

---

> > ### Comment · Reviewer_BvMn · 2022-08-09
> > **Thank you for the responses**
> >
> > I would like to thank the authors for the response. However, my concerns about limited technical novelty are not addressed.
> >
> > In the response, the authors listed four references to show that PCA is widely studied, yet three of them are published at least 12 years ago. The only recent reference tackles the missing not at random problem, which is important and challenging. This submission, however, directly applies the standard PCA method to some fully observed features. Even if other features have missing values, they do not affect the PCA part at all. The only added value brought by this submission seems to be the speedup by using PCA, which could be obtained by any other existing dimensionality reduction methods like SVD and LDA. The way of using PCA to speed up imputation as proposed in this submission is quite straightforward to me.
> >
> > From an engineering perspective, I agree that this paper does some good work in examining the proposed method, but in terms of methodology, I still think that the contribution of this paper is too weak to be accepted by NeurIPS.

---

> > > ### Author Response · Authors · 2022-08-09
> > > **Response to Reviewer [BvMn]**
> > >
> > > Thank you very much for your comments. We agree that SVD and LDA have great potential to be applied in a similar fashion to speed up imputation, since LDA is a popular supervised dimension reduction technique that makes use of the information from the labels, and  SVD is another commonly used technique for sparse data. Therefore, we think it is worth studying how they can be applied to speed up imputation extensively for a supervised learning problem/for problems with sparse data, in the future. In this revision, we have explored how different formulations of PCA can be applied to speed up imputation in a simple fashion that can be used for continuous/categorical/mixed data and can work with many imputation techniques, including state-of-the-art ones.

---

### Meta-Review · Area_Chair_gr8G · 2022-08-26

**Recommendation:** Reject
**Confidence:** Certain

**Metareview:**

This paper proposes a framework based on principle components analysis (PCA) to speed up the missing data imputation. It divides the feature sets into two partitions -- the fully observed one and the one that contains missing values. The proposed method applies PCA to the fully observed partition to do dimensionality reduction, followed by the existing imputation methods. The authors further propose to apply PCA to the imputed data to speed up the downstream classification task.

The major weakness is that the methodological contribution is quite limited. Projecting data into lower dimensional spaces to speed up downstream tasks is not new. In particular, the main assumption of random missingness has been considered before 10-20 years ago and the more challenging setting of non-random missingness was not considered. Overall the reviewers mostly agree that the contribution is limited.

**Award:**

No

---

### Decision · Program_Chairs · 2022-09-14

Reject